# Generating Differentially Private Datasets Using GANs

## Abstract

In this paper, we present a technique for generating artificial datasets that retain statistical properties of the real data while providing differential privacy guarantees with respect to this data. We include a Gaussian noise layer in the discriminator of a generative adversarial network to make the output and the gradients differentially private with respect to the training data, and then use the generator component to synthesise privacy-preserving artificial dataset. Our experiments show that under a reasonably small privacy budget we are able to generate data of high quality and successfully train machine learning models on this artificial data.

## 1 Introduction

Following recent advancements in deep learning (Silver et al., 2016; He et al., 2015; Wu et al., 2016), more and more people and companies are interested in putting their data in use as they see that machine learning is able to generate a wide range of benefits, including financial, social, medical, security, and so on. At the same time, however, such models are often able to capture a fine level of detail in training data potentially compromising privacy of individuals who's features sharply differ from others. This problem is partially mitigated by the use of regularisation techniques that "smooth out" outstanding details and avoid overfitting, but it does not give any theoretical privacy guarantees. Recent research by Fredrikson et al. (2015) suggests that even without access to internal model parameters, by using hill climbing on output probabilities of a neural network, it is possible to recover (up to a certain degree) individual faces from a training set.

The latter result is especially disturbing knowing that deep learning models are becoming an integral part of our lives, making its way to phones, smart watches, cars, and appliances. And since these models are often trained on customers data, such training set recovery techniques will endanger privacy even without access to the manufacturer's servers where these models are being trained.

In order to protect privacy while still benefiting from the use of statistics and machine learning, a number of techniques for data anonymisation has been developed over the years, including $k$-anonymity (Sweeney, 2002), $l$-diversity (Machanavajjhala et al., 2007), $t$-closeness (Li et al., 2007), and differential privacy (Dwork, 2006; Dwork et al., 2006; Dwork, 2008; Dwork et al., 2014). The latter has been recognised as a strong standard and is widely accepted by the research community.

We study the task of publishing datasets in a differentially private manner. In particular, we are interested in solving two problems. First, we want to be able to benefit from the use of machine learning by third parties while protecting sensitive information of individuals in our dataset. Second, we want to be sure that even if adversaries get access to the third-party model trained on our data, they would not be able to recover private information. An additional challenge is to be able to publish an entire dataset, as opposed to being required to use a query interface like in a typical differentially private framework.

In this paper, we propose a simple solution to this problem. The main idea of our approach is to use generative adversarial networks (GANs) introduced in Goodfellow et al. (2014), trained with addition of Gaussian noise in the embedding space, to create artificial datasets that follow the same distribution as the real data while providing differential privacy guarantees. This method has a number of advantages over the methods proposed earlier. First of all, this solution is simple to implement, e.g. it does not require training ensembles of models on disjoint data. Second, it can be done on a user side, and not on the side of the machine learning service provider, which eliminates

the necessity of trusting this service provider or implementing privacy-preserving models locally. Third, similarly to Abadi et al. (2016), privacy cannot be compromised even if the entire trained model is accessible to an adversary.

Our contributions in this paper are the following:

- we propose a novel mechanism for non-interactive differentially private data release, and to the best of our knowledge this is the first practical solution for complex real-world data;
- we introduce a new technique of preserving privacy in neural networks via adding noise in the forward pass during training;
- we show that this technique guarantees differential privacy for both the outputs and the learned weights of the network;
- we demonstrate that we are able to achieve high accuracy in learning tasks while maintaining a reasonable (single-digit) privacy budget.

The remainder of the paper is structured as follows. In Section 2, we give an overview of related work. Section 3 contains necessary background on differential privacy and generative adversarial networks. In Section 4, we describe our approach and provide its theoretical analysis and some practical aspects. Experimental results and implementation details are presented in Section 5, and Section 6 concludes the paper. The theorem proofs and additional details can be found in the Appendix.

## 2 RELATED WORK

Given the level of attention to deep learning and the rising importance of privacy, it is unsurprising that there has been a significant increase in the number of publications on the topic of privacy-preserving deep learning (and machine learning in general) in recent years.

One take on the problem is to distribute training and use disjoint sets of training data. An example of such approach is the paper of Shokri & Shmatikov (2015), where they propose to train in a distributed manner by communicating sanitised updates from participants to a central authority. Such a method, however, yields high privacy losses as pointed out by Abadi et al. (2016) and Papernot et al. (2016). An alternative technique, also using disjoint training sets, suggested by Papernot et al. (2016), applies an ensemble of independently trained teacher models and semi-supervised knowledge transfer to a student model to achieve almost state-of-the-art (non-private) accuracy on MNIST (LeCun et al., 1998) and SVHN (Netzer et al., 2011) with single-digit differential privacy bounds. This work was based on a paper by Hamm et al. (2016) and extends their method to generic learning models with any type of loss functions or optimisation algorithms. To the best of our knowledge, this is the most accurate privacy-preserving learning result to date, although one has to make sure that all the teaching ensemble and the aggregator are inaccessible to an adversary and the model is queried for teachers' votes only a small number of times.

A somewhat different approach is taken in Abadi et al. (2016). They suggest using differentially private stochastic gradient descent (for brevity, we will refer to it as DP-SGD in the remainder of the paper) to train deep learning models in a private manner. This approach allows to achieve high accuracy while maintaining low differential privacy bounds, and does not require distributed training.

As stated above, our goal is to enable data usage by third party machine learning service providers to benefit from their expertise. All of the aforementioned methods, however, require every provider of such service to comply with the chosen privacy-preserving procedure which is not realistic. An alternative solution to this problem is to focus on sanitising data and making sure that training machine learning models on it would not compromise privacy. This direction is taken, for example, by Bindschaedler et al. (2017). The authors use a graphical probabilistic model to learn an underlying data distribution and transform real data points (seeds) into synthetic data points. Synthetic data is then filtered by a privacy test based on a plausible deniability criterion, which can be equivalent to differential privacy under certain conditions.

Our approach, on the other hand, is to generate private data without requiring any real seeds. Thus, there is no need for privacy tests at the release stage, and the only requirement is that the generative

model is privacy-preserving. By using GANs (Goodfellow et al., 2014) we ensure that our method is scalable and applicable to complex real-world data.

## 3 BACKGROUND

This section gives a short introduction to GANs and differential privacy. Another important notion is the moments accountant method (Abadi et al., 2016) used to compute actual privacy bounds during training. However, since it is not essential for understanding the paper, we defer its description to the Appendix.

### 3.1 GENERATIVE ADVERSARIAL NETWORKS

In recent years, generative adversarial networks (Goodfellow et al., 2014; Salimans et al., 2016) and its extensions, such as DCGAN (Radford et al., 2015) and EBGAN (Zhao et al., 2016), have received great attention and pushed the boundaries for deep generative models along with variational autoencoders (VAEs) (Kingma & Welling, 2014; Rezende et al., 2014; Gregor et al., 2015) and recursive neural networks (e.g. PixelRNN by Oord et al. (2016)). The most successful application for such generative models so far has been realistic image generation, perhaps due to abundance of training data and inherent geometric structure.

In our work, we decided to choose GANs for several reasons. Firstly, GANs have shown very good results in practice, generating sharper images compared to other generative models. Secondly, the forward pass for generating data is much faster than that of, for instance, RNNs. Thirdly, the *generator* part of the model, the one we eventually interested in, does not interact with real training data at any point in the learning process, only getting gradients from the *discriminator*.

In short, GANs can be described as follows. The model consists of two separate components: the *generator* $\mathcal{G}(z)$ and the *discriminator* $\mathcal{D}(x)$. The generator's goal is to produce realistic samples of data based on a random variable $z \sim p_z(z)$, while the discriminator is tasked with distinguishing real data samples $x \sim p_{\text{data}}(x)$ from generated samples $\hat{x} \sim p_g(x)$. These two models are trained in an adversarial fashion, essentially playing a two-player game, with the goal to converge to the Nash equilibrium. Since training GANs in practice can be challenging, there is a number of commonly used tricks to improve convergence, such as using the Adam optimisation method (Kingma & Ba, 2015), feature matching, batch normalisation, and one-sided label smoothing (Salimans et al., 2016). We also observe improvements with adding labels to the discriminator (Odena, 2016) and unrolling discriminator updates (Metz et al., 2016).

### 3.2 DIFFERENTIAL PRIVACY

The notion of differential privacy has been introduced and extended in a series of papers by Dwork et al. (Dwork, 2006; Dwork et al., 2006; Dwork, 2008; Dwork et al., 2014), and is regarded as a strong privacy standard. It is defined for two adjacent datasets that differ by a single element:

**Definition 1.** *A randomized mechanism $\mathcal{M} : \mathcal{D} \to \mathcal{R}$ with domain $\mathcal{D}$ and range $\mathcal{R}$ satisfies $(\varepsilon, \delta)$-differential privacy if for any two adjacent inputs $d, d' \in \mathcal{D}$ and for any subset of outputs $S \subseteq \mathcal{R}$ it holds that:*

$$\Pr\left[\mathcal{M}(d) \in S\right] \le e^{\varepsilon} \Pr\left[\mathcal{M}(d') \in S\right] + \delta \tag{1}$$

Among the mechanisms to achieve differential privacy, two of the most widely used are Laplacian and Gaussian noise mechanisms. We are primarily interested in the latter, because of the improved privacy bounds analysis provided by the moments accountant method described in the Appendix. The Gaussian noise mechanism is defined as follows:

$$\mathcal{M}(d) \triangleq f(d) + \mathcal{N}(0, s_f^2 \cdot \sigma^2), \tag{2}$$

where $s_f$ is the sensitivity of $f$ (i.e. $s_f = |f(d) - f(d')|$ for $f : \mathcal{D} \to \mathbb{R}$), and $\mathcal{N}(0, s_f^2 \cdot \sigma^2)$ is the Gaussian distribution with the mean 0 and the standard deviation $s_f \sigma$.

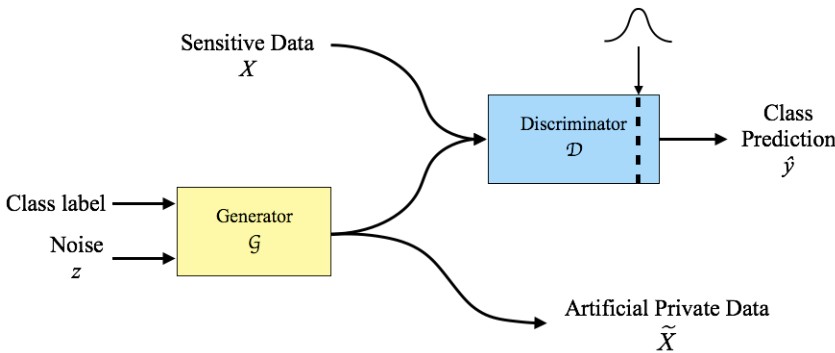

Figure 1: Architecture of our solution. Sensitive data $X$ is fed into a discriminator $\mathcal{D}$ with a privacy-preserving layer (dashed line). This discriminator is used to train a differentially private generator $\mathcal{G}$ to produce a private artificial dataset $\widetilde{X}$.

## 4 OUR APPROACH

In this section, we describe our solution and provide a theoretical proof of privacy guarantees, as well as discuss limitations of the method. Let us begin with the formal problem statement.

**Problem Statement.** *Given the dataset $X \sim p_{\text{data}}(x)$, generate an artificial dataset $\widetilde{X} = \mathcal{M}(X)$ using the privacy mechanism $\mathcal{M} : \mathbb{X} \to \mathbb{X}$, such that*

1. *it follows the same data distribution: $\widetilde{X} \sim p_{\text{data}}(x)$;*

2. *it provides differential privacy guarantees: $\Pr\left[\mathcal{M}(X) \in \mathbb{S}\right] \leq e^{\varepsilon} \Pr\left[\mathcal{M}(X') \in \mathbb{S}\right] + \delta$ for any adjacent datasets $X, X'$, and for any $\mathbb{S} \subseteq \mathbb{X}$.*

Here $\mathbb{X} = \{X \mid X \sim p_{\text{data}}(x)\}$ is the space of all datasets formed by points drawn from the same distribution $p_{\text{data}}(x)$.

In most real-world problems, the true data distribution $p_{\text{data}}(x)$ is unknown and needs to be estimated empirically. Since we are primarily interested in data synthesis, we will turn to generative models, and in particular we are going to use GANs as the mechanism to estimate $p_{\text{data}}(x)$ and draw samples from it. If trained properly, GAN will provide a solution to the sub-problem (1).

Despite the fact that the generator does not have access to the real data $X$ in the training process, one cannot guarantee differential privacy because of the information passed through with the gradients from the discriminator. A simple high level example will illustrate such breach of privacy. Let the datasets $X, X'$ contain small real numbers. The only difference between these two datasets is the number $x' \in X'$, which happens to be extremely large. Since the gradients of the model depend on $x'$, one of the updates of the discriminator trained on $X'$ may be very different from the rest, and this difference will the be propagated to the generator breaking privacy in general case.

In order to maintain differential privacy guarantees, we propose the following solution.

**Proposition.** *Introduce a Gaussian noise layer in the discriminator network of GAN, so that its output, and therefore the weights of the trained generator, are differentially private with respect to the input data $X$. Use this generator to create a publishable differentially private dataset.*

The components of our solution are depicted in Figure 1.

### 4.1 THEORETICAL ANALYSIS OF THE APPROACH

To validate the proposed solution, we first analyse it theoretically and show that the addition of a Gaussian noise layer in the discriminator network yields differential privacy in the generator. We will take the following steps to do that:

1. analyse privacy of the output of the noise layer w.r.t. the inputs $X$ and $X'$;

2. determine privacy bounds on the output of the whole network;

3. show that the same bounds hold for gradient updates.

Let us start by describing the setting and notation used in the remainder of the section. We are given two adjacent datasets $(X, y)$ and $(X', y')$ and a deterministic feed-forward neural network $\mathcal{N}$ with a Gaussian noise layer $\pi$. We denote the inputs of the layer $\pi$ as $x_\pi$ and $x'_\pi$, and the outputs of the final layer of the network $\hat{y} = \mathcal{N}(X)$ and $\hat{y}' = \mathcal{N}(X)$ correspondingly. To ensure $(\varepsilon, \delta)$-differential privacy of $\pi$, the standard deviation of the noise has to be at least $\sigma = C\sqrt{2\log(1.25/\delta)}/\varepsilon$, where $C$ is the sensitivity of the preceding layer's output $x_\pi$.

**Lemma 1.** *If the output of the noise layer $\pi(x_\pi)$ is $(\varepsilon, \delta)$-differentially private w.r.t. $x_\pi$ and the network layers before $\pi$ preserve adjacency of $X$ and $X'$, then $\pi(X)$ is also $(\varepsilon, \delta)$-differentially private w.r.t. $X$.*

The proof of this lemma and the following Theorems 1 and 2 can be found in the appendix.

Using Lemma 1, we are able demonstrate that the outputs of a feed-forward neural network with a Gaussian noise layer are differentially private with respect to the input data, which is expressed in the following theorem.

**Theorem 1. (Forward pass)** *The output $\hat{y}$ of a deterministic feed-forward neural network $\mathcal{N}$ with $(\varepsilon, \delta)$-differentially private layer $\pi$, is also $(\varepsilon, \delta)$-differentially private with respect to $X$.*

Now, given that the forward pass is differentially private, we can formulate the main theoretical result of the paper: differential privacy of the gradients, and thus, the weights of the network $\mathcal{N}$.

**Theorem 2. (Backward pass)** *Given a feed-forward neural network $\mathcal{N}$ with $(\varepsilon, \delta)$-differentially private outputs $\hat{y}$, weight updates $\omega_X^{(i)}$ are also $(\varepsilon, \delta)$-differentially private with respect to $X$ in each iteration $i$ of gradient descent.*

Since we are interested in generating data using GANs, we will also need the following corollary to finalise the theoretical foundation for our framework.

**Corollary 1. (GANs)** *Given a generative adversarial network consisting of the generator $\mathcal{G}$ and the discriminator $\mathcal{D}$ with a privacy-preserving layer, gradient updates of $\mathcal{G}$ will have the same privacy bounds as gradient updates of $\mathcal{D}$.*

*Proof.* This result trivially follows from Theorem 2 once we observe that generator updates are a function of discriminator updates. □

The above analysis is applicable for each individual iteration of the gradient descent, and privacy bounds on the final parameters can be obtained using composition theorems or a more efficient moments accountant method (Abadi et al., 2016).

Note that Theorems 1 and 2 define differential privacy of the neural network with respect to the inputs $X$ only, not taking into account the labels $y$. In certain cases, when labels of interest are already a public knowledge and do not reveal any information about data, it may be sufficient. However, if labels privacy is required, it is possible to incorporate it in the proposed approach in two ways.

A first solution is to modify the learning problem so that labels become a part of data. For example, if one wants to train a face recognition model with privacy-breaking labels (e.g. specific names—John, Bob, Julia, etc.), it is possible to add these labels to $X$, and instead use *True* and *False* labels in $y$, indicating whether the input image and the input name correspond to each other. This way, label privacy will be handled by the same framework.

Alternatively, one can use a separate privacy-preserving mechanism to retrieve labels during training. In this case, the eventual privacy w.r.t. the pair $(X, y)$ may be derived from a composition of two mechanisms, which is shown in the theorem below. One possible candidate for such mechanism is the noisy voting scheme as used in Papernot et al. (2016).

**Theorem 3. (Private labels)** *Given a feed-forward neural network $\mathcal{N}$ with $(\varepsilon_1, \delta_1)$–differentially private outputs $\hat{y}$, and the training labels $\widetilde{y}$ satisfying $(\varepsilon_2, \delta_2)$–differential privacy w.r.t. the true*

*labels $y$, the gradient updates $\omega_X^{(i)}$ are $(\varepsilon_1 + \varepsilon_2, \delta_1 + \delta_2)$–differentially private with respect to $(X, y)$ on each iteration $i$ of gradient descent.*

*Proof.* There are two privacy mechanisms $\mathcal{M}_1$ and $\mathcal{M}_2$ applied to $X$ and $y$ correspondingly. Observe that $\mathcal{M}_1$ does not have access to $y$, and thus, $y$ cannot influence the output probabilities of $\mathcal{M}_1$. The same is true for $\mathcal{M}_2$ and $X$. Consequently, we can assume that both mechanisms are applied to a pair $(X, y)$. This allows us to employ a basic sequential composition theorem for differential privacy (Dwork & Lei, 2009) to obtain the privacy bounds. □

While it may appeal to use parallel composition instead of sequential composition to obtain a tighter bound, since $X$ and $y$ appear to be disjoint, it would be incorrect. The reason is that $X$ and $y$ are strongly correlated and breaking privacy of one can reveal the other. Alternatively, one could use advanced composition theorems (see e.g. Dwork et al. (2010); Kairouz et al. (2017)) to prove tighter privacy bounds, but it is not the goal of our paper.

### 4.2 PRACTICAL ASPECTS

Based on the analysis above, we can do a number of important observations regarding applicability of this technique.

First of all, the analysis is performed for feed-forward networks. Other architectures, such as RNNs, LSTMs, or memory networks, require additional investigation. Second, we focused on deterministic networks, meaning that the only two sources of stochasticity are data shuffling and privacy-preserving noise layer $\pi$. Additional randomness in the network would complicate the proofs by introducing uncertainty in mappings. Third, conditions of Lemma 1 dictate that the network layers prior to $\pi$ must preserve adjacency of the input. One layer breaking this condition is batch normalisation, because it introduces interdependencies between examples inside a batch, and just one different instance can change an entire batch.

Summarising these limitations, the neural network under question must

- be a feed-forward network;
- not have randomised layers, e.g. dropout;
- not have adjacency breaking layers before the privacy layer, e.g. batch normalisation.

In the following section, we will touch upon some implications of it that affect practical performance. Note that these restrictions only apply to the network, in which we insert a privacy-preserving layer, i.e. only the discriminator in our case.

## 5 EVALUATION

In this section, we provide some implementation details and discuss evaluation results obtained on MNIST (LeCun et al., 1998) and SVHN (Netzer et al., 2011) datasets.

### 5.1 EXPERIMENTAL SETUP

We evaluate our solution as follows. First, we train a generative model on original datasets (using only training parts of each) with differential privacy by adding a Gaussian noise layer to the discriminator. We will call this model *a teacher*, analogously to Papernot et al. (2016). Then, we generate an artificial dataset of comparable size using the obtained model. Finally, we train a separate (non-private) classifier, which we call *a student*, on generated data and test it using held-out test sets. The last step is important from two perspectives: we can quantify the quality of generated samples as opposed to visual inspection typically done with GANs, and we can compare test errors to previously reported values. Note that there is no dependencies between the teacher and the student models. Moreover, student models are not constrained to neural networks and can be implemented as any type of machine learning algorithm.

We choose two commonly used image classification datasets for our experiments: MNIST and SVHN. MNIST is a handwritten digit recognition dataset consisting of 60'000 training examples

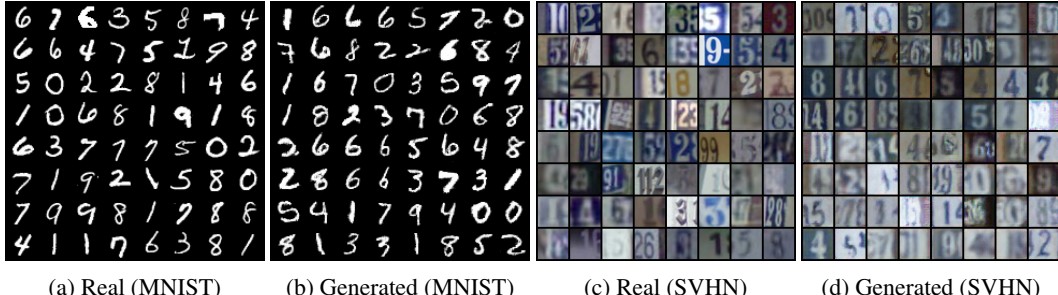

| (a) Real (MNIST) | (b) Generated (MNIST) | (c) Real (SVHN) | (d) Generated (SVHN) |

Figure 2: Real and generated examples for MNIST and SVHN dataset.

Table 1: Accuracy of student models for given privacy bounds for our method and semi-supervised knowledge transfer approach of Papernot et al. (2016). In both cases, we restricted our method to have tighter privacy bounds.

| Dataset | $\varepsilon$ | $\delta$ | Non-Private Baseline | Papernot et al. (2016) | Our approach |
|---------|-----|-----|----------------------|------------------------|--------------|
| MNIST | $\leq 8$ | $10^{-5}$ | 99.18% | 98.10% | 98.19% |
| SVHN | $\approx 8$ | $10^{-6}$ | 92.80% | 90.66% | 83.49% |

and 10'000 test examples, each example is a 28x28 size greyscale image. SVHN is also a digit recognition task, with 73'257 images for training and 26'032 for testing. The examples are coloured 32x32 pixel images of house numbers from Google Street View.

## 5.2 IMPLEMENTATION DETAILS

Implementation was done in Python using Pytorch[1]. For generative model, we used a modified version of DCGAN by Radford et al. (2015). More specifically, the discriminator consists of five (four for MNIST) convolutional layers followed by leaky ReLU activations and a linear classifier with sigmoid output. We clip the output of the third convolutional layer (to ensure bounded sensitivity) and add Gaussian noise before passing it to the remaining convolutions with batch normalisation. The generator has two linear layers in front of five deconvolutions with batch normalisation and ReLU activations, ensued by fractional max pooling with tanh activation at the end.

Both networks were trained using Adam optimiser (Kingma & Ba, 2015) with parameters typical for GAN training: learning rate set to 0.0002, $\beta_1 = 0.5$, $\beta_2 = 0.999$, and a batch size of 32. Privacy bounds were evaluated using the moments accountant and the privacy amplification theorem (Abadi et al., 2016), and therefore, are data-dependent and are tighter than using normal composition theorems.

The student network is constructed of two convolutional layers with ReLU activations, batch normalisation and max pooling, followed by two fully connected layers with ReLU, and a softmax output layer. Again, training is performed by Adam algorithm. It is worth mentioning that this network does not achieve state-of-the-art performance on the used datasets, but we are primarily interested in evaluating the performance drop compared to a non-private model rather than getting the best test score.

## 5.3 DISCUSSION

Using the experimental setup and implementation described above, we were able to get results close to Papernot et al. (2016) although not quite matching their accuracy for the same privacy bounds on SVHN. A performance gap is expected due to more generic nature of our method and a simpler privacy-preserving procedure. Overall, we managed to achieve 98.19% accuracy on MNIST and 83.49% accuracy on SVHN while maintaining approximately $(3.45, 10^{-5})$ and $(8, 10^{-6})$-differential privacy. These numbers, along with the corresponding results of Papernot et al. (2016),

---

[1]http://pytorch.org

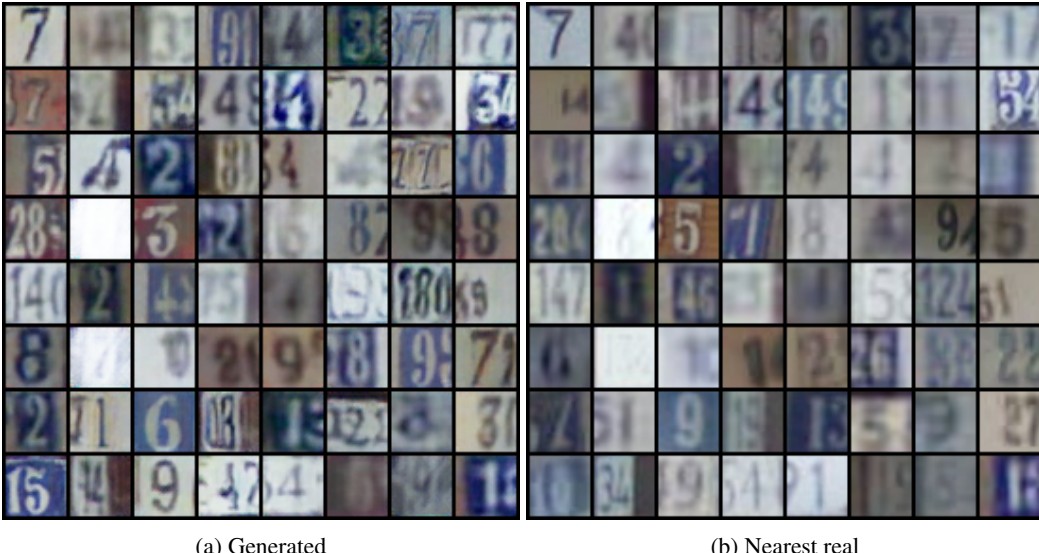

(a) Generated          (b) Nearest real

Figure 3: Generated examples and nearest neighbours from real data (SVHN).

can be found in Table 1. It is also worth noting that we did not perform rigorous hyper-parameter tuning due to limited computational resources; even better accuracy could be achieved have we had done that. Additionally, we trained a simple logistic regression model on MNIST, and obtained 88.96% accuracy on privately generated data compared to 92.58% on the original data, which confirms that any model can be used as a student.

Examples of real and generated privacy-preserving images for MNIST and SVHN data are depicted on Figure 2. It can be seen that generated images don't have the same contrast and dynamic range as real examples, which is not a problem in non-private GANs. We attribute it to the lack of batch normalisation in the discriminator.

In addition to quantitative analysis of test errors and privacy bounds, we perform visual inspection of generated examples and corresponding nearest neighbours in real data. Figure 3 depicts a set of generated private examples and their nearest real counterparts. We observe that while some generated images are very close to real examples they don't match exactly, differing either in shape, colour or surrounding digits. Moreover, a lot of pairs come from entirely different classes.

## 6  CONCLUSIONS

We investigate the problem of non-interactive private data release with differential privacy guarantees. We employ generative adversarial networks to produce artificial privacy-preserving datasets. Contrary to existing privacy protection work in deep learning, this method allows to publish sanitised data and train any non-private models on it. The choice of GANs as a generative model ensures scalability and makes the technique suitable for real-world data with complex structure. Moreover, this method does not require running privacy tests on generated data before releasing it.

Additionally, we introduce a novel method for preserving privacy of training data specific to deep neural networks based on adding noise in the embedding space during forward pass. It provides differential privacy guarantees and allows to construct privacy-preserving models in a simple and straightforward fashion, without modifying optimisation algorithms.

In our experiments, we show that student models trained on artificial data can achieve high utility on MNIST dataset, while maintaining performance costs of added privacy and flexibility at acceptable levels on a more complicated SVHN data. Adding privacy directly to the trained model still provides better accuracy, and therefore, one of the possible directions for future work is to improve the quality of generated data for given privacy bounds. Extending presented technique and analysis to other types of deep neural networks provides another exciting opportunity for further research.

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

## 7    APPENDIX

In this appendix, we state again and prove lemmas and theorems from Section 4.1.

### 7.1    PROOF OF LEMMA 1

**Lemma 2.** *If the output of the noise layer $\pi(x_\pi)$ is $(\varepsilon, \delta)$-differentially private w.r.t. $x_\pi$ and the network layers before $\pi$ preserve adjacency of $X$ and $X'$, then $\pi(X)$ is also $(\varepsilon, \delta)$-differentially private w.r.t. $X$.*

*Proof.* By definition of differential privacy:

$$P[\pi(x_\pi) \in \mathbb{S}] \le e^\varepsilon P[\pi(x'_\pi) \in \mathbb{S}] + \delta, \tag{3}$$

for all adjacent $x_\pi$ and $x'_\pi$.

We need to show that the same holds for all adjacent inputs $X, X'$, i.e. $P[\pi(X) \in \mathbb{S}] \le e^\varepsilon P[\pi(X') \in \mathbb{S}] + \delta$. Observe that we defined our network as deterministic (i.e. not having any randomness apart from initial data shuffling). Therefore, $P[X_\pi | X] = \delta_{x_\pi}(X_\pi)$, where $\delta_x(X)$ is a Dirac delta function. Conceptually, it means that the entire mass of the distribution of $X_\pi$ is concentrated on the point $x_\pi$.

Using the above observation,

$$P[\pi(X) \in \mathbb{S}] = \int_{X_\pi} P[\pi(X_\pi) \in \mathbb{S}] P[X_\pi | X] \, dX_\pi \tag{4}$$

$$= \int_{X_\pi} P[\pi(X_\pi) \in \mathbb{S}] \delta_{x_\pi}(X_\pi) \, dX_\pi \tag{5}$$

$$= P[\pi(x_\pi) \in \mathbb{S}] \tag{6}$$

$$\le e^\varepsilon P[\pi(x'_\pi) \in \mathbb{S}] + \delta \tag{7}$$

$$= \int_{X_\pi} \left( e^\varepsilon P[\pi(X_\pi) \in \mathbb{S}] + \delta \right) \delta_{x'_\pi}(X_\pi) \, dX_\pi \tag{8}$$

$$= \int_{X_\pi} \left( e^\varepsilon P[\pi(X_\pi) \in \mathbb{S}] + \delta \right) P[X_\pi | X'] \, dX_\pi \tag{9}$$

$$= \le e^\varepsilon P[\pi(X') \in \mathbb{S}] + \delta \tag{10}$$

$\square$

**Remark.** *Allowing randomised layers in the network would complicate the proof due to marginalisation over all possible outcomes $X_\pi$ corresponding to the input $X$.*

### 7.2    PROOF OF THEOREM 1

**Theorem 1. (Forward pass)** The output $\hat{y}$ of a deterministic feed-forward neural network $\mathcal{N}$ with $(\varepsilon, \delta)$-differentially private layer $\pi$, is also $(\varepsilon, \delta)$-differentially private with respect to $X$.

*Proof.* Using the lemma above, we can show that outputs of the layer $\pi$ are $(\varepsilon, \delta)$-differentially private w.r.t. the inputs $X$, i.e.

$$P[\pi(X) \in \mathbb{S}] \le e^\varepsilon P[\pi(X') \in \mathbb{S}] + \delta \tag{11}$$

Since we require all the layers of $\mathcal{N}$ (except $\pi$) to be deterministic, there is a deterministic mapping from the outputs of $\pi$ to $\hat{y}$. Let us denote this mapping $f(\pi)$, and the preimage of a set $\mathbb{S}$ under this mapping $f^{-1}[\mathbb{S}]$ (i.e. $f^{-1}[\mathbb{S}] = \{\pi : f(\pi) \in \mathbb{S}\}$).

Note that we treat $X$ and $X'$ as points in the space of all datasets $\mathcal{X}$, and thus, $\pi$ and $f$ are not set-valued functions. Also, to avoid confusion, let us restate that $f^{-1}[\mathbb{S}]$ is a *preimage* of a set $\mathbb{S}$ under $f$, and not a function inverse. Hence, we do not require $f$ to be bijective, or even injective.

Using the above,

$$P[\hat{y} \in \mathbb{S}] = P[f(\pi(X)) \in \mathbb{S}] \tag{12}$$

$$= P[\pi(X) \in f^{-1}[\mathbb{S}]] \tag{13}$$

$$\leq e^{\varepsilon} P[\pi(X') \in f^{-1}[\mathbb{S}]] + \delta \tag{14}$$

$$= e^{\varepsilon} P[f(\pi(X')) \in \mathbb{S}] + \delta \tag{15}$$

$$= e^{\varepsilon} P[\hat{y}' \in \mathbb{S}] + \delta, \tag{16}$$

for any pair of adjacent datasets $X$ and $X'$ (differing in one training example), thus, proving the theorem. $\qquad\square$

## 7.3   PROOF OF THEOREM 2

**Theorem 2.  (Backward pass)** Given a feed-forward neural network $\mathcal{N}$ with $(\varepsilon, \delta)$-differentially private outputs $\hat{y}$, weight updates $\omega_X^{(i)}$ are also $(\varepsilon, \delta)$-differentially private with respect to $X$ in each iteration $i$ of gradient descent.

*Proof.* Let us denote by $g(y, \hat{y}) = \frac{\partial \mathcal{L}(y, \hat{y})}{\partial \omega}$ the gradient of the loss function w.r.t. network parameters. Similarly to Theorem 1, the preimage of a set $\mathbb{T}$ under $g$ is denoted by $g^{-1}[y, \mathbb{T}] = \{\hat{y} : g(y, \hat{y}) \in \mathbb{T}\}$. To better connect it with Theorem 1 let us define $\mathbb{S} = g^{-1}[y, \mathbb{T}]$.

Since gradient is a function of network outputs and labels, we have

$$P[g(y, \hat{y}) \in \mathbb{T}] = P[\hat{y} \in g^{-1}[y, \mathbb{T}]] = P[\hat{y} \in \mathbb{S}]. \tag{17}$$

Combining the above results,

$$P[\omega_X^{(i)} \in \mathbb{T}] = P[g(y, \hat{y}) \in \mathbb{T}] \tag{18}$$

$$= P[\hat{y} \in \mathbb{S}] \tag{19}$$

$$\leq e^{\varepsilon} P[\hat{y}' \in \mathbb{S}] + \delta \tag{20}$$

$$= e^{\varepsilon} P[g(y, \hat{y}') \in \mathbb{T}] + \delta \tag{21}$$

$$= e^{\varepsilon} P[\omega_{X'}^{(i)} \in \mathbb{T}] + \delta, \tag{22}$$

for any pair of adjacent datasets $X$ and $X'$, demonstrating that weight updates stay $(\varepsilon, \delta)$-differentially private w.r.t to the input. $\qquad\square$

## 7.4   MOMENTS ACCOUNTANT

The privacy bound produced by the strong composition theorem is often too loose, and therefore, we exploit the moments accountant technique developed by Abadi et al. (2016) for analysing their DP-SGD algorithm.

To give the main idea of the method, let us start with defining the *privacy loss*.

**Definition 2.** *Let $\mathcal{M} : \mathcal{D} \rightarrow \mathcal{R}$ be a randomized mechanism and $d, d'$ a pair of adjacent databases. Let $\mathtt{aux}$ denote an auxiliary input. For an outcome $o \in \mathcal{R}$, the privacy loss at $o$ is defined as:*

$$c(o; \mathcal{M}, \mathtt{aux}, d, d') \triangleq \log \frac{\Pr[\mathcal{M}(\mathtt{aux}, d) = o]}{\Pr[\mathcal{M}(\mathtt{aux}, d') = o]}. \tag{23}$$

*And the privacy loss random variable $C(\mathcal{M}, \mathtt{aux}, d, d')$ is defined as $c(\mathcal{M}(d); \mathcal{M}, \mathtt{aux}, d, d')$.*

The moments accountant is then defined as follows:

**Definition 3.** *Again, let $\mathcal{M} : \mathcal{D} \rightarrow \mathcal{R}$ be a randomized mechanism, $d, d'$ a pair of adjacent databases, and $\mathtt{aux}$ denote an auxiliary input. The moments accountant is*

$$\alpha_{\mathcal{M}}(\lambda) \triangleq \max_{\mathtt{aux}, d, d'} \alpha_{\mathcal{M}}(\lambda; \mathtt{aux}, d, d'), \tag{24}$$

*where $\alpha_{\mathcal{M}}(\lambda; \mathtt{aux}, d, d') \triangleq \log \mathbb{E}[exp(\lambda C(\mathcal{M}, \mathtt{aux}, d, d'))]$ is a moment-generating function.*

In short, the moments accountant method tracks the bounds on the moments of the privacy loss random variable and then uses Markov inequality to obtain the tail bound on this random variable corresponding to the values of $\varepsilon$ and $\delta$.

