# OpenReview forum: "Generating Differentially Private Datasets Using GANs"
_ICLR.cc/2018/Conference — Reject_

### Official Review · AnonReviewer2 · 2017-11-26
**I like the paper because of its central idea, and the importance of the problem. However, I am not confident about the technical novelty in the algorithmic components.**

**Rating:** 6
**Confidence:** 4

**Review:**

Summary: The paper addresses the problem of non-interactive differentially private mechanism via adversarial networks. Non-interactive mechanisms have been one of the most sought-after approaches in differentially private algorithm design. The reason is that once a differentially private data set is released, it can be used in any way to answer queries / perform learning tasks without worrying about the privacy budget. However, designing effective non-interactive mechanisms are notoriously hard because of strong computational lower bounds. In that respect, the problem addressed in this paper is extremely important, and the approach of using an adversarial network for the task is very natural (yet novel).

The main idea in the paper is to set up a usual adversarial framework with the generator and the discriminator, where the discriminator has access to the raw data. The information (in the form of gradients) is passed from the discriminator on to the generator via a differentially private channel (using Gaussian mechanism).

Positive aspects of the paper: One main positive aspect of the paper is that it comes up with a very simple yet effective approach for a non-interactive mechanism for differential privacy. Another positive aspect of the paper is that it is very well-written and is easy to follow.

Questions: I have a few questions about the paper.

1. The technical novelty of the paper is not that high. Given the main idea of using a GAN, the algorithms and the experiments are fairly straightforward. I may be missing something. I believe the paper can be strengthened by placing more emphasis on the technical content.

2. I am mildly concerned about the effectiveness of the algorithm in the high dimensional setting. The norm of i.i.d. Gaussian noise scales roughly as \sqrt{dimensions}, which may be too much to tolerate in most settings.

3. I was wondering if there is a way to incorporate assumptions about sparsity in the original data set, to handle curse of dimensionality.

4. I am not sure about the novelty of Theorem 2. Isn't it just post-processing property of differential privacy?

---

### Official Review · AnonReviewer3 · 2017-11-27
**May need more details for privacy analysis**

**Rating:** 5
**Confidence:** 4

**Review:**

The paper proposes a technique for differentially privately generating synthetic data using GAN, and experimentally showed that their method achieves both high utility and good privacy.
The idea of building a differentially private GAN and generating differentially private synthetic data is very interesting. However, my main concern is the privacy aspect of the technique, as it is not explained clearly enough in the paper. There is also room for improvement in the presentation and clarity of the paper.

More details:
- About the differential privacy aspect:
  The author didn't provide detailed privacy analysis of the Gaussian noise layer, and I don't find the values of the sensitivity (C = 1) provided in the answer to a public comment easy to see. Also, the paper mentioned that the batch size is 32 and the author mentioned in the comment that the std of the Gaussian noise is 0.7, and the number of epoch is 50 or 150. I think these values would lead to epsilon much larger than 8 (as in Table 1). However, in Section 5.2, it is said that "Privacy bounds were evaluated using the moments accountant and the privacy amplification theorem (Abadi et al., 2016), and therefore, are data-dependent and are tighter than using normal composition theorems." I don't see clearly why privacy amplification is needed here, and why using moments accountant and privacy amplification can lead to data-dependent privacy loss.
  In general, I don't find the privacy analysis of this paper clear and detailed enough to convince me about the correctness of the privacy results. However, I am very happy to change my opinion if there are convincing details in the rebuttal.

- About the presentation:
  As a paper proposing a differentially private algorithm, detailed and formal analysis of the privacy guarantees is essential to convince the readers. For example, I think it would be much better if there is a formal theorem showing the sensitivity of the Gaussian noise layer. And it would be better to restate (in Appendix 7.4) not only the definition of moments accountant, but the composition and tail bound, as well as the moments accountant for the Gaussian mechanism, since they are all used in the privacy analysis of this paper.

---

### Official Review · AnonReviewer1 · 2017-11-29
**Ok, but not good enough**

**Rating:** 4
**Confidence:** 4

**Review:**

This paper considers the problem of generating differentially private datasets using GANs. To the best of my knowledge this is the first paper to study differential privacy for GANs.

The paper is fairly well-written but has several major weaknesses:
-- Privacy parameter eps = 8 used in the experiments implies that the likelihood of any event can change by e^8 which is roughly 3000, which is an unacceptably high privacy loss. Moreover, even for this high privacy loss the accuracy on the SVHN dataset seems to drop a lot (92% down to 83%) when proposed mechanism is used.
-- I didn't find a formal proof of the privacy guarantee in the paper. The authors say that the privacy guarantee is based on the moments accountant method, but I couldn't find the proof anywhere. The method itself is introduced in Section 7.4 but isn't used for the proof. Thus the paper seems to be incomplete.

---

### Public Comment · (anonymous) · 2017-11-13
**Privacy parameters**

Could you please specify which values have you used in your experiments for the following parameters?
1) C - sensitivity of the preceding layer’s output
2) number of training epochs
3) magnitude of Gaussian noise (i.e, std deviation) injected per batch iteration

Thanks in advance

---

> ### Author Response · Authors · 2017-11-15
> **Privacy parameters**
>
> Thank you for your question. We use the following parameter values in our experiments:
> 1). C = 1, in all of the experiments.
> 2). Number of training epochs for GAN is 150 for SVHN and 50 for MNIST. Note that we also use unrolling of the discriminator for 4 steps (reduced to 3 steps after 120 epochs) in generator updates to avoid mode collapse.
> 3). Standard deviation of the Gaussian noise is generally set to 0.7. On SVHN, it is increased to 0.8 after 120 epochs to meet a tighter privacy bound.

---

### Public Comment · (anonymous) · 2017-11-29
**Dimension of the data**

Hi, interesting work. But the noise is added during each iteration and that would end up to be large. Did you run the algorithm for high dimension data?
What's the difference between the proposed method and the one in this paper? https://www.biorxiv.org/content/biorxiv/early/2017/07/05/159756.full.pdf

---

> ### Author Response · Authors · 2017-12-01
> **Dimension of the data**
>
> Thank you for your questions.
> To answer the first question, adding noise in each iteration is not a problem, as it does not introduce bias over time. The dimensionality of data would not be an issue either, because noise is added in an embedding space (and not in the original feature space) for each dimension independently, making the method agnostic to the dimensionality of original data. As reported in the paper, we have done experiments with the SVHN dataset, which has 3072-dimensional input vectors.
> Moving on to your second question. Thank you for drawing our attention to this paper. We were not aware of it and missed it when studying the related work.
> While the main ideas are indeed similar, there is a major difference in our method: the way of preserving privacy in GAN training. On the initial stages of our work, we explored the possibility of using differentially private SGD, but we found that achieving reasonable privacy bounds requires adding too much noise to gradients and makes GAN training much harder than it already is. The aforementioned paper confirms our findings by showing that the noise quickly overpowers the gradient (Fig. 1(e)) and that using the GAN after the final epoch is not sufficient for obtaining realistic data (Fig. 2(a)). Instead, we propose adding noise in the forward pass, which improves convergence properties and generated data quality.
> This difference leads to a number of advantages. Most importantly, our technique does not require additional procedures for picking specific generator epochs or modifying optimisation methods. Moreover, it can be implemented by simply adding a noise layer to the discriminator, and we formally show that this is sufficient for achieving differential privacy.

---

### Public Comment · (anonymous) · 2017-12-28
**About necessary of generator**

Dear authors,

The topic of this paper is interesting, but I have the following question: the authors showed that deterministic feed-forward neural network with (\epsilon, \delta)-differentially private layer is (\epsilon, \delta)-differentially private in Theorem 1 and 2. Therefore, it seems that there is no need to use "generator" for privacy since deterministic feed-forward neural network with noise layer already provides privacy guarantees. In other words, the reason why the authors introduced generator is not clear...

If I don't understand the paper correctly, please do not hesitate to let me know.

Thanks in advance.

---

### Author Response · Authors · 2018-01-01
**Dimensionality problem**


Dear readers,

We have discovered a problem related to dimensionality that invalidates privacy guarantees stated in the paper. We are currently working on solving the issue.

---

### Decision · Program_Chairs · 2018-01-29
**ICLR 2018 Conference Acceptance Decision**

**Decision:**

Reject

**Comment:**

This paper presents an interesting idea: employ GANs in a manner that guarantees the generation of differentially private data.

The reviewers liked the motivation but identified various issues. Also, the authors themselves discovered some problems in their formulation; on behalf of the community, thanks for letting the readers know.

The discovered issues will need to be reviewed in a future submission.